# *Pratylenchus* *vovlasi* sp. Nov. (Nematoda: Pratylenchidae) on Raspberries in North Italy with a Morphometrical and Molecular Characterization [note 1]

**DOI:** 10.3390/plants10061068

**Published:** 2021-05-26

**Authors:** Alberto Troccoli, Elena Fanelli, Pablo Castillo, Gracia Liébanas, Alba Cotroneo, Francesca De Luca

**Affiliations:** 1Consiglio Nazionale delle Ricerche (CNR), Istituto per la Protezione Sostenibile delle Piante (IPSP), v. G. Amendola, 122/D, 70126 Bari, Italy; alberto.troccoli@ipsp.cnr.it (A.T.); elena.fanelli@ipsp.cnr.it (E.F.); 2Institute for Sustainable Agriculture (IAS), Spanish National Research Council (CSIC), Avda. Menéndez Pidal s/n, 14004 Córdoba, Spain; p.castillo@csic.es; 3Departamento de Biología Animal, Biología Vegetal y Ecología, Campus Las Lagunillas, Universidad de Jaén, 23071 Jaén, Spain; gtorres@ujaen.es; 4Regione Piemonte-Settore Fitosanitario e Servizi Tecnico-Scientifici, 10144 Torino, Italy; albaccotroneo@gmail.com

**Keywords:** Bayesian inference, D2-D3 expansion domains of the 28S rRNA gene, *hsp90* gene, integrative taxonomy, ITS

## Abstract

Root-lesion nematode species rank third only to root-knot and cyst nematodes as having the greatest economic impact on crops worldwide. A survey of plant-parasitic nematodes associated with decaying raspberries (*Rubus* sp.) in northern Italy revealed that root-lesion nematodes were the most frequently occurring species among other phytonematodes. Several *Pratylenchus* species have been associated with *Rubus* sp. in Canada (Quebec, British Columbia) and USA (North Carolina, Maryland, New Jersey) including *P. penetrans* and *P. crenatus.* In the roots and rhizosphere of symptomatic raspberries, nematodes of two *Pratylenchus* spp. were detected. Detailed morphometrics of the two root-lesion nematode isolates were consistent with *Pratylenchus crenatus* and with an undescribed *Pratylenchus* species. The extracted nematodes were observed and measured as live and fixed materials and subsequently identified by integrative taxonomy (morphometrically and molecularly). The latter species is described herein as *Pratylenchus vovlasi* sp. nov., resulting morphometrically closest to *P. mediterraneus* and phylogenetically to *P. pratensis*. The molecular identification of *Pratylenchus vovlasi* sp. nov. was carried out by sequencing the ITS region, D2-D3 expansion domains of the 28S rRNA gene and a partial region of the nuclear *hsp90* gene. ITS-RFLP and sequence analyses revealed that *Pratylenchus vovlasi* sp. nov. had species-specific restriction profiles with no corresponding sequences present in the database. The phylogenetic relationships with ITS and D2-D3 sequences placed the *Pratylenchus vovlasi* sp. nov. in a clade with *P. pratensis* and *P. pseudopratensis*. This research confirms the occurrence of cryptic biodiversity within the genus *Pratylenchus* as well as the need for an integrative approach to the identification of *Pratylenchus* species.

## 1. Introduction

Raspberries (*Rubus* sp.) have a long history of human consumption and cultivation in Europe, with Russia being the leading producer [1]. Raspberries have been eaten fresh for thousands of years. In the last decades, there has been an increase in the demand for fruit and fruit-based products as consumers seek out healthier dietary options. In particular, berries are considered one of the best dietary sources of bioactive compounds that have important antioxidant properties, with associated health effects such as protective effects against several cancers and cardiovascular disorders [2,3] and thus the berry market is expected to expand during the next years. Raspberries are the most productive in areas with mild winters and long, moderate summers. The production of raspberries is greatly influenced by biotic and abiotic factors. Among the biotic factors, plant-parasitic nematodes have an important role in the reduction of the raspberry yield [4,5]. Several genera of plant-parasitic nematodes have been found to be associated with the raspberry [4,5,6,7,8,9]. The most frequently observed species in soil and the roots of raspberries belong to the root-lesion nematode genus *Pratylenchus*, mainly *P. crenatus* Loof, 1960, (71.7% of soil samples, 53.4% of root samples), *P. penetrans* (Cobb, 1917) Filipjev and Schuurmans Stekhoven, 1941 (51.6%, 54.2%) and *P. scribneri* Steiner in Sherbakoff and Stanley 1943 (9.2%, 10.2%; [9]). In Italy, raspberries are grown mainly in the northern areas including Trentino Alto Adige, the main production area, Verona province and Piedmont. Other Italian areas growing raspberries are in Romagna in the north and Calabria, Sicily, Campania and Basilicata in the south. Considering the economic significance of root-lesion nematodes on raspberries and the need to accurately distinguish these damaging species for their practical management in the field, we provide here the morphometrical and molecular characterization of a new root-lesion nematode, *Pratylenchus vovlasi* sp. nov. parasitizing raspberries in the Piedmont area. The specific objectives of this paper were: (i) to carry out a comprehensive identification with morphological and morphometric approaches of *P. vovlasi* sp. nov. with a differential diagnosis to closely related species; (ii) to provide a molecular characterization of *P. vovlasi* sp. nov. and estimate its phylogenetic relationships with other representatives of the *Pratylenchus* genus using three molecular markers: two ribosomal markers (the D2-D3 expansion segments of the 28S rRNA and the internal transcribed spacer region (ITS rRNA) and the nuclear region of the partial heat shock protein 90 (*hsp*90) gene.

## 2. Results

Two species were identified from raspberries in the Piedmont region. Detailed morphometrics and molecular analyses of the two root-lesion nematode populations were consistent with *P. crenatus* and with an undescribed *Pratylenchus* species. As *P. crenatus* is a well-known and well molecular characterized species, we concentrated our efforts on the integrative taxonomic identification of the undescribed species.

### 2.1. Molecular Characterization

For molecular analyses, a total of five individual specimens for each molecular marker were amplified. The D2-D3 expansion domains of *28S* rRNA, *ITS* rRNA and partial *hsp90* genes of the new species *P. vovlasi* sp. nov. yielded single fragments of ~800 bp, 700 bp and 300 bp, respectively, based on a gel electrophoresis. The D2-D3 for *P. vovlasi* sp. nov. (OA984892–OA984893) showed a very low intraspecific variability with one different nucleotide and no indels (99% similarity). The D2-D3 for *P. vovlasi* sp. nov. differed from the closest related species, *P. pseudopratensis* Seinhorst, 1968 (JX261965) by 18 nucleotides and 0 indels (97% similarity), *P. pratensis* (De Man, 1880) Filipjev, 1936 (KY828298) by 23–26 nucleotides and 0 indels (96–97% similarity) and from *P. vulnus* (JQ003993) by 72–73 nucleotides and four to ten indels (89–90% similarity).

The ITS region for *P. vovlasi* sp. nov. also showed a low intraspecific variability by 19 nucleotides and 6 indels (97% similarity). The ITS1 for *P. vovlasi* sp. nov. (OA984869–OA984870) showed a low similarity with all of the ITS sequences of *Pratylenchus* spp. deposited in NCBI including the most similar species, *P. pratensis* (=*P. lentis*) (AM933158, AM933147, AM933149) and *P. fallax* Seinhorst, 1968, (FJ719921, FJ719917), by 97–98 different nucleotides and 34–39 indels (86% similarity).

The partial *hsp*90 gene amplified products of two individual specimens were cloned and four clones for each individual specimen were sequenced (OA984937–OA984944). The sequence analyses revealed the occurrence of two different fragments for each specimen and both fragments coded for *hsp90* differed in the length of the intron (52 bp vs. 43 bp) and nucleotide variability. The 298 bp fragment showed an 82% similarity (254/310 identities) with the 310 bp fragment at a nucleotide level. At an amino acid level, the two fragments showed 96% identities (81/84) and 98% positives (83/84). The new *hsp90* sequences for *P. vovlasi* sp. nov. showed a high intraspecific variability by 3–60 nucleotides and 0–13 indels (81–99% similarity). The *hsp90* sequences for *P. vovlasi* sp. nov. differed by 13, 16 and 61 nucleotides and 4, 4 and 24 indels from the most closely related species, *P. speijeri* De Luca, Troccoli, Duncan, Subbotin, Waeyenberge, Coyne, Brentu and Inserra, 2012 (HE601547) with a 96% similarity, *P. coffeae* (Zimmermann, 1898) Filipjev and Schuurmans Stekhoven, 1941 (HE601548) with a 95% similarity and *P. hippeastri* Inserra, Troccoli, Gozel, Bernard, Dunn and Duncan, 2007 (HE601549) with an 81% similarity, respectively.

### 2.2. Restriction Profiles

PCR-RFLP analyses of the ITS region allowed us to determine the species-specific patterns for the Italian population of *P. vovlasi* sp. nov. (Figure 1) that clearly identified this species.

### 2.3. Phylogenetic Relationships

The phylogenetic relationships among *Pratylenchus* species inferred from the analyses of the D2-D3 expansion domains of 28S rRNA, ITS and the partial *hsp90* gene sequences using BI are shown in Figure 2, Figure 3 and Figure 4, respectively. The phylogenetic trees generated with the ribosomal and nuclear markers included 54, 60 and 31 sequences with 693, 548 and 284 positions in length, respectively (Figure 2, Figure 3 and Figure 4). The D2-D3 tree of *Pratylenchus* spp. showed a well-supported subclade (PP = 1.00) including *P. vovlasi* sp. nov., *P. pratensis* and *P. pseudopratensis* (Figure 2). All other clades followed the same pattern as previous studies on *Pratylenchus* species.

The 50% majority-rule consensus ITS BI tree showed several clades that were not well-defined (Figure 2) but a well-supported subclade (PP = 1.00) including *P. vovlasi* sp. nov. and *P. pratensis* (Figure 3). This subclade was phylogenetically related to *P. vulnus* Allen and Jensen, 1951 and *P. kumamotoensis* Lal and Khan, 1990 in a moderately supported subclade (PP = 0.95) (Figure 3). Finally, the *hsp90* BI tree confirmed the occurrence of two isoforms within *P. vovlasi* sp. nov. that were clearly separated in two independent well-supported subclades (PP = 0.98 and PP = 1.00, respectively) (Figure 4).

### 2.4. Morphology and Morphometry of Pratylenchus vovlasi sp. nov.

***Pratylenchus vovlasi* sp. nov.** (http://zoobank.org/urn:lsid:zoobank.org:act:3B91A2A8-B809-47BA-A822-CF380EBC0245, accessed on 2 April 2021).

#### 2.4.1. Description

Female: the body assumes an almost straight to open C posture when heat-killed (Figure 5A). The lip region is slightly offset from body contour and bears three annuli, which narrow in diameter towards the anterior end (Figure 5D and Figure 6D). In the en face SEM view lip region, they appear dumb-bell-shaped (Figure 7A,B,D,I) with an acute pattern (sensu Subbotin et al. [10]) fitting the group II according to the classification scheme of Corbett and Clark [11]. The stylet is relatively small and delicate with conus about 45 ± 2.5 (41–49) % of the entire stylet length (Table 1). The stylet shaft is slender ending with rounded basal knobs, which are slightly anteriorly flattened. The pharyngeal procorpus narrows just anterior to the small, oval metacorpus. The valve of the median bulb is conspicuous. The isthmus is short, encircled by a nerve ring and widening to a pharyngeal lobe with a dorsal nucleus just posterior to the cardia and ventro-sublateral nuclei near the tip of the pharyngeal lobe (Figure 5B,C) and overlapping the intestine ventro-laterally for almost a two-body diameter at the cardia level. The secretory-excretory pore is level with the cardia just posterior to the hemizonid. The body annulation is distinct; the lateral field usually with four smooth incisures and a few specimens have an additional line (Figure 6K) or more rarely a few oblique striae (Figure 6J) in the middle of the central band. The outline of outer bands becomes indented towards the tail end, posterior to phasmid, with the inner lines fusing just posterior to the phasmid. The genital tract is well developed with oocytes arranged in a single row. The spermatheca is large and spherical to oval, usually full of sperm (Figure 5F,E,G); its posterior margin is 50.0 ± 9.6 (36.5–64.5) µm from the vagina (Table 1). The vulva is slightly sunken and the vulval lips are not prominent (Figure 5E,F). The post-uterine sac is about 1.1 vulval body diameter long and is usually undifferentiated. The phasmids are located to the mid-tail, 13.2 ± 2.2 (11–17) µm from the tail tip. The tail is typically subcylindrical, tapering towards the tip with a rounded-truncate terminus in most specimens; a few specimens have broadly rounded, conical tails with coarsely pointed or indented striated termini (Figure 5H,I and Figure 6F,H,I,L).

Male: similar to the female except in the posterior end of the body and in a slightly smaller and usually slender body length. The lip region is usually higher and narrower than in the female (Figure 6B,C). The stylet is slightly smaller than that of female with smaller, more rounded knobs. The pharyngeal bulb is small and round; the isthmus is slender and rather short ending in a long, narrow glandular lobe. The testis is outstretched and filled with round spermatozoa in the vas deferens. The spicules are paired, weakly cephalated and ventrally arcuate. The gubernaculum is simple and slightly curved. The tail is conical and bent on the ventral side with a prominent, crenate bursa.

#### 2.4.2. Type Host and Locality

*Pratylenchus vovlasi* sp. nov. was found associated with the roots and soil of *Rubus* sp. with a population density of 650 nematodes/250 cm^3^ of soil in the locality of Prarostino, Turin province, Piedmont Region, north Italy.

#### 2.4.3. Type Material

Holotype, female and male paratypes mounted on glass slides were deposited in the nematode collection at the Istituto per la Protezione Sostenibile delle Piante (IPSP), CNR, Bari, Italy (collection numbers IPSP-M-1238-1250). The additional paratypes were distributed to the United States Department of Agriculture Nematode Collection, Beltsville, MD, USA (collection number IPSP-M-1237), WaNeCo Plant Protection Service, Wageningen, The Netherlands, (collection number IPSP-M-1246) and Instituto de Agricultura Sostenible, CSIC, Córdoba, Spain (collection number IPSP-M-1245).

#### 2.4.4. Diagnosis and Relationships

*Pratylenchus vovlasi* sp. nov. is characterized by a lip region slightly offset with three annuli narrowing towards the anterior end, sub-median sectors fused with the oral disc and separated by a lateral sector to give a dumb-bell -shaped pattern; the stylet is rather small (15.0 µm long) with rounded knobs and a short pharyngeal overlap. The lateral field has four smooth incisures in most specimens, the spermatheca is round to oval and is usually full of sperm, the vulva is located in a relative anterior position and the tail is subcylindrical with a truncate, smooth terminus and with common males. The matrix code of the new species according to Castillo and Vovlas [12] is A2 B2, C2, D2, E2, F3, G1,2, H1, I2, J1, K1 (Table 2).

Related species sharing with *P. vovlasi* sp. nov. a three lip annuli, a divided face as seen with SEM (Group 2 according to Corbett and Clark [11]), a functional spermatheca and numerous males include *P. bhattii* Siddiqi, Dabur and Bajaj, 1991, *P. kralli* Ryss, 1982, *P. mediterraneus* Corbett, 1983, *P. thornei* Sher and Allen, 1953, *P. pratensis*, *P. pseudopratensis* Seinhorst, 1968, *P. penetrans*, *P. fallax* and *P. convallariae* Seinhorst, 1959.

*P. vovlasi* sp. nov. is most closely related to *P. mediterraneus*, matching 10 out of 11 characteristics according to the matrix code [12] and differing from it just by a slightly shorter pharyngeal overlap (32 ± 6.7 (20–43) vs. 25–55 μm, code I2 vs. I3 in the tabular key), a face pattern showing transverse incisures in the middle of the sub-medial sectors (not present in *P. mediterraneus*), lateral sectors fused (vs. separated) with the oral disc, smooth lateral fields, not areolated vs. crenate (outer bands), occasionally areolated and with the middle band variously ornamented (Figure 6J,K and Figure 7K). From *P. pratensis*, *P. fallax*, *P. penetrans* and *P. vulnus*, it differs in the en face SEM pattern (belonging to Group 2 vs. Group 3, according to Corbett and Clark, [11]) and the tail tip morphology despite a certain degree of overlap (mostly truncate and smooth vs. usually oblique and annulated in *P. pratensis*, rounded or with a slightly irregular contour in *P. fallax*, generally rounded in *P. penetrans* and narrowly rounded to subacute and occasionally irregular in *P. vulnus*). Furthermore, the new species differs from *P. pratensis* by a mostly rounded vs. oval to rectangular spermatheca and the number of tail annuli (14–20 vs. 20–28); from *P. fallax* by a shorter stylet (range: 14.3–16.3 vs. 16–17 μm) and a smaller c’ value 1.8 ± 0.2 (1.4–2.3) vs. 2.5 (2.0–3.0); from *P. penetrans* by a slightly shorter stylet mean length (15 vs. 16), a slightly more anterior position of the vulva (78 (74–80) vs. 78–84%) and in a fewer number of tail annuli on the ventral surface (15.6 ± 2.0 (14–20) vs. 15–27); from *P. vulnus* by a shorter post-uterine sac (1.1 vs. ca. 2.0 vulval body diameter; code F3 vs. F6 after Castillo and Vovlas, [12] (Table 2) and a more anterior position of the vulva (78 (74–80) vs. 77–82%). From *P. pseudopratensis*, it differs in the en face SEM view (not dumb-bell-shaped in *P. pseudopratensis*), a shorter stylet (range: 14.3–16.3 vs. 16–17 μm), the shape of the spermatheca (rectangular, sometimes empty in *P. pseudopratensis*) and the tail tip (smooth vs. crenated; code H1 vs. H2 after Castillo and Vovlas, [12]).

Other *Pratylenchus* species with three lip annuli, a functional spermatheca and the presence of males and with a matrix code (sensu Castillo and Vovlas, [12]) similar to *P. vovlasi* sp. nov. have been described without examining their lip patterns. These species include *P. convallariae*, *P. kralli* and *P. bhattii*, from which the new species differs by the following characteristics: from *P. convallariae*, by a shorter stylet (15 (14.3–16.3), feature C2 in Table 2, vs. 17 (16–18) μm feature C3), a shorter PUS (1.1, feature F3 vs. more than 1.4–2 vulval body diameter, feature F6 in Table 2) and by a smooth tail tip (H1) vs. coarsely and often irregularly annulated (H2); from *P. kralli*, in the stylet knob shape (mostly rounded vs. anteriorly directed), the post-uterine sac length (1.1 vs. more than 1.5 vulval body diameter), the tail tip is truncated, smooth and more rarely pointed vs. pointed and showing a slight groove; from *P. bhattii* by a slightly longer stylet (15 (14.3–16.3) vs. 13.5 (13–14) μm) and a more posterior vulva (V = 78 vs. 73% mean value) with the vulval lips continuous with the body contour vs. raised in a prominent protuberance. Finally, the new species can be compared with *P. rwandae* Singh, Nyiragatare, Janssen, Couvreur, Decraemer and Bert, 2018, but differs from it by a labial region in the en face view showing clearly separated sub-median sectors with transverse incisures in the middle (vs. slightly separated from the lateral sectors and without transverse incisures), males present (vs. absent), females with mostly round and full spermatheca (vs. oval to rounded and empty), lateral fields with four incisures (vs. six or more at the mid-body) and a tail with a fewer number of ventral annuli (14–20) and usually a truncated tip (vs. 18–28 tail annuli and a highly variable tail tip).

#### 2.4.5. Etymology

The species epithet, *P. vovlasi*, is dedicated to Dr. Nikos Vovlas, an eminent Italian nematologist and taxonomist from the Istituto per la Protezione Sostenibile delle Piante (IPSP), Consiglio Nazionale delle Ricerche (CNR), Bari, Italy.

## 3. Discussion

The identification of the *Pratylenchus* species is difficult because many diagnostic characteristics overlap and also due to the increasing number of nominal species. An accurate identification is needed in order to adopt appropriate control strategies. The primary objective of this study was to identify and molecularly characterize root-lesion nematodes parasitizing raspberries cultivated in the Piedmont region. The present study reports on the occurrence of two root-lesion nematodes associated with raspberry fields in the Piedmont region, *P. crenatus*, along with an abundant species herein identified as *P. vovlasi* sp. nov. *Pratylenchus crenatus* was previously reported on raspberries in several European countries [6,9].

Our results demonstrated that the application of rRNA molecular markers integrated with morphological studies could help in the diagnosis and characterization of root-lesion nematode species. Based on the molecular characterization using the D2-D3 expansion domains of the 28S rRNA gene, ITS region and the partial hsp90 gene, the abundant species was clearly identified as *P. vovlasi* sp. nov. This species proved very similar in morphometry and morphology to *P. mediterraneus*, differing only by a shorter pharyngeal overlap, as well as with *P. thornei*, *P. penetrans*, *P. fallax* and *P. convallariae*. By blasting at NCBI the D2-D3 region, it was 96–97% similar to *P. pratensis* and *P. pseudopratensis*, respectively, while by using ITS sequences it was 85–87% similar to *P. pratensis* and *P. fallax*. These results represented an additional confirmation of the extraordinary cryptic diversity of the nematodes of the genus *Pratylenchus*, as reported in previous studies [13,14,15,16,17]. Phylogenetic analyses of the ITS and the D2-D3 sequences confirmed a sister relationship between *P. vovlasi* sp. nov. with *P. pratensis* and *P. pseudopratensis*. Furthermore, in both phylogenetic trees, the new species was closely related to *P. vulnus* and *P. kumamotoensis* (Figure 2 and Figure 3). Major clades for D2-D3 and ITS phylogenetic trees were highly correlated with previous phylogenetic studies carried out by Subbotin et al. [10], Palomares-Rius et al. [14,18] and Araya et al. [17]. We agreed with De Luca et al. [13] who suggested that the ITS-containing region allowed a better discrimination among the closely related species studied because it evolved faster than the D2-D3 expansion segments of 28S rDNA and accumulated more substitution changes.

The current study confirmed the occurrence of different hsp90 isoforms in the *Pratylenchus* species as already reported by Fanelli et al. [19]. In *P. vovlasi* sp. nov. the different isoforms differed from each other in the length of the intron and the nucleotide variability grouping in two well-supported clusters. This finding suggested that the different isoforms of *P. vovlasi* sp. nov. hsp90 arose by gene duplication events relatively recently because the nucleotide variability was low and the gene structure was still conserved [20,21]. Similarly, these results confirmed that the primers for the amplification of the hsp90 gene could also amplify other paralogous genes in Pratylenchus and could be used for species delimitation within this genus [22,23]. Furthermore, the occurrence of different *hsp90* isoforms in *P. vovlasi* sp. nov. confirmed that this gene family could contribute to the adaptation to different hosts and to different environments.

In this study, *P. vovlasi* sp. nov. was isolated from raspberries in north Italy together with *P. crenatus* in the same area. Further research on its pathogenicity and the economic damage on this crop is needed.

## 4. Materials and Methods

### 4.1. Nematode Isolate and Morphological Studies

No specific permits, other than that of the farm owner, were required for the indicated fieldwork studies. The soil samples were obtained in raspberry cultivated areas in Piedmont and did not involve any endangered species or those protected in Italy, nor were the sites protected in any way.

The soil samples recovered from the rhizosphere and roots of the raspberries located in the Piedmont region were sent to the IPSP laboratory, Bari, in the autumn of 2010. The samples were collected with a shovel from the upper 50 cm of the soil of raspberries arbitrarily chosen at random. The nematodes were extracted from 500 cm^3^ of the soil by centrifugal flotation [24]. The specimens for light microscopy were killed by gentle heat, fixed in a solution of 4% formaldehyde + 1% propionic acid and processed to pure glycerin using Seinhorst’s method [25]. The specimens were examined with a Zeiss III compound microscope with a Nomarski differential interference contrast at powers up to ×1000. The measurements and drawings were made at the camera lucida on glycerin-infiltrated specimens. All measurements were expressed in micrometers (µm) unless otherwise stated.

For the scanning electron microscope (SEM), fixed specimens were dehydrated in a gradient series of ethanol, critical-point dried, sputter-coated with gold according to the protocol of Abolafia et al. [26] and observed with a Zeiss Merlin Scanning Electron Microscope (5 kV; Zeiss, Oberkochen, Germany).

### 4.2. DNA Extraction, Polymerase Chain Reaction (PCR) and Sequencing

DNA was extracted from 20 single individual root-lesion nematode specimens. The specimens were handpicked and placed singly on a glass slide in 3 µL of the lysis buffer (10 mM Tris-HCl, pH 8.8, 50 mM KCl, 15 mM MgCl_2_, 0.1% Triton × 100, 0.01% gelatin with 90 µg/mL proteinase K) and then cut into small pieces by using a sterilized syringe needle under a dissecting microscope. The samples were incubated at 65 °C for 1 h and then at 95 °C for 15 min to deactivate the proteinase K [27]. The following sets of primers were used for the amplification of the gene fragments in the present study: (i) D2-D3 expansion segments of the 28S rRNA gene using forward D2A and reverse D3B primers; (ii) ITS1-5.8-ITS2 rRNA using forward TW81 and reverse AB28 primers [28,29]; (iii) the *hsp*90 gene using forward U831 and reverse L1110 primers [30]. New sequences were submitted to the GenBank database under the accession numbers indicated on the phylogenetic trees.

### 4.3. PCR-RFLP

Ten μL of each ITS-PCR product from three individual nematodes were digested with five units of the following restriction enzymes: *Alu*I, *Ava*II, *Bam*HI, *Dde*I, *Hin*fI and *Rsa*I. The digested products were separated onto a 2.5% agarose gel by electrophoresis, stained with gel red dye, visualized on a UV transilluminator and photographed with a digital system.

### 4.4. Phylogenetic Analysis

Sequenced genetic markers in the present study (after discarding primer sequences and ambiguously aligned regions) and several *Pratylenchus* spp. sequences obtained from GenBank were used for the phylogenetic reconstruction. The outgroup taxa for each dataset were selected based on previous published studies [17,18,23]. Multiple sequence alignments of the newly obtained and published sequences were made using the FFT-NS-2 algorithm of MAFFT v. 7.450 [31]. The sequence alignments were visualized using BioEdit [32] and edited by Gblocks ver. 0.91b [33] in the Castresana Laboratory server (http://molevol.cmima.csic.es/castresana/Gblocks_server.html, accessed on 2 April 2021) using options for a less stringent selection (minimum number of sequences for a conserved or a flanking position: 50% of the number of sequences + 1; maximum number of contiguous no conserved positions: 8; minimum length of a block: 5; allowed gap positions: with half).

The phylogenetic analyses of the sequence datasets were based on Bayesian inference (BI) using MRBAYES 3.2.7a [34]. The best-fit model of DNA evolution was calculated with the Akaike information criterion (AIC) of JMODELTEST v. 2.1.7 [35]. The best-fit model, the base frequency, the proportion of invariable sites and the gamma distribution shape parameters and substitution rates in the AIC were then used in the phylogenetic analyses. The BI analyses were performed under a general time-reversible model with a proportion of invariable sites and a rate of variation across sites (GTR + I + G) model for D2-D3 and the partial *hsp90* gene and under a transversional model with a proportion of invariable sites and a rate of variation across sites (TVM + I + G) model for the ITS region. These BI analyses were run separately per dataset with four chains for 4 × 10^6^ generations. The Markov chains were sampled at intervals of 100 generations. Two runs were conducted for each analysis. After discarding the burn-in samples of 30% and evaluating the convergence, the remaining samples were retained for more in-depth analyses. The topologies were used to generate a 50% majority-rule consensus tree. Posterior probabilities (PP) were given on appropriate clades. The trees from all analyses were visualized using FigTree software version v.1.42 [36].

## Figures and Tables

**Figure 1 plants-10-01068-f001:**
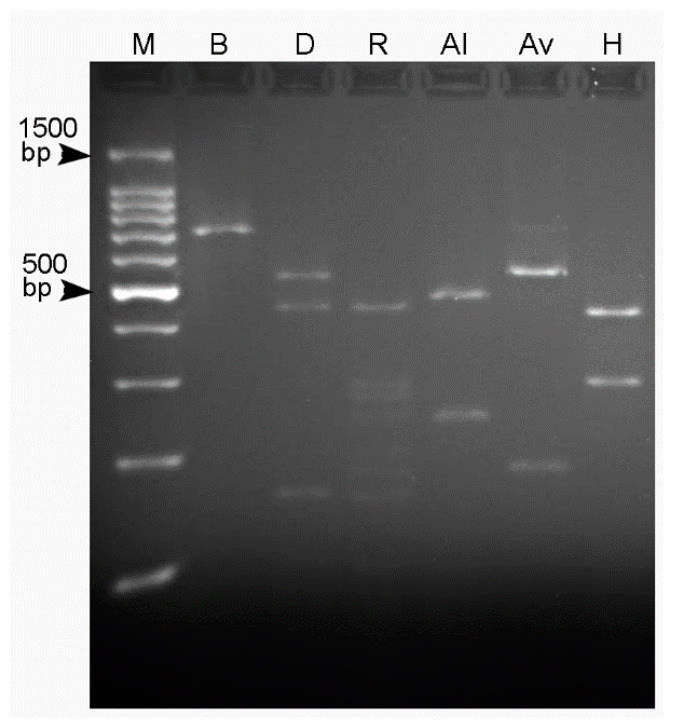
Restriction fragments of an amplified ITS of *Pratylenchus vovlasi* sp. nov. Al: *Alu*I, Av: *Ava*II, B: *Bam*HI, D: *Dde*I, H: *Hinf*I, R: *Rsa*I and M: 100 bp ladder.

**Figure 2 plants-10-01068-f002:**
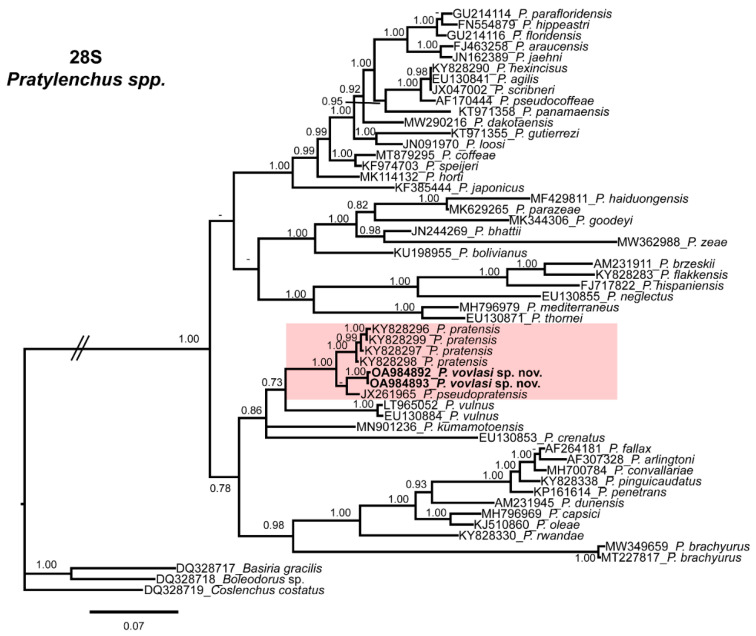
Phylogenetic relationships of *Pratylenchus vovlasi* sp. nov. within the genus *Pratylenchus*. A Bayesian 50% majority-rule consensus tree as inferred from the D2 and D3 expansion domains of the 28S rRNA sequence alignment under the general time-reversible model of the sequence evolution with a correction for invariable sites and a gamma-shaped distribution (GTR + I + G). Posterior probabilities more than 0.70 are given for appropriate clades. Newly obtained sequences in this study are shown in bold. Scale bar = expected changes per site.

**Figure 3 plants-10-01068-f003:**
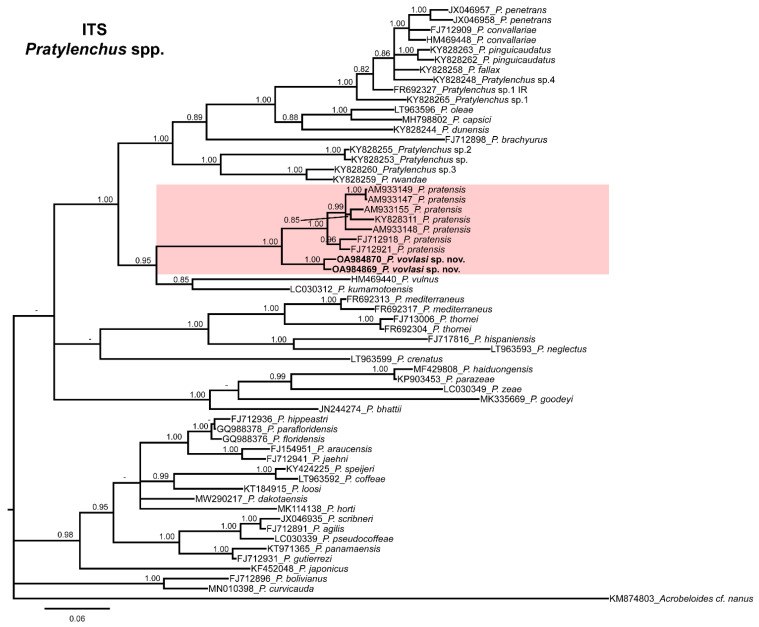
Phylogenetic relationships of *Pratylenchus vovlasi* sp. nov. within the genus *Pratylenchus*. A Bayesian 50% majority-rule consensus tree as inferred from the ITS sequence alignment under a transversional model with a proportion of invariable sites and a rate of variation across sites (TVM + I + G). Posterior probabilities more than 70% are given for appropriate clades. Newly obtained sequences in this study are in bold letters. Scale bar = expected changes per site.

**Figure 4 plants-10-01068-f004:**
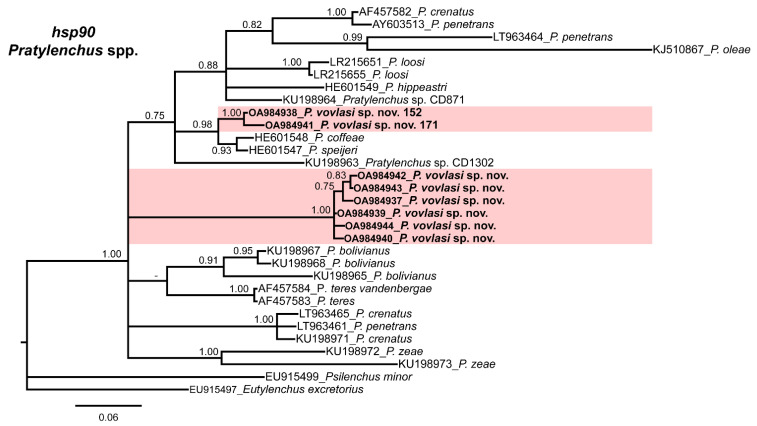
Phylogenetic relationships of *Pratylenchus vovlasi* sp. nov. within the genus *Pratylenchus*. A Bayesian 50% majority-rule consensus tree as inferred from the *hsp90* sequence alignment under the general time-reversible model of the sequence evolution with a correction for invariable sites and a gamma-shaped distribution (GTR + I + G). Posterior probabilities more than 70% are given for appropriate clades. Newly obtained sequences in this study are in bold letters. Scale bar = expected changes per site.

**Figure 5 plants-10-01068-f005:**
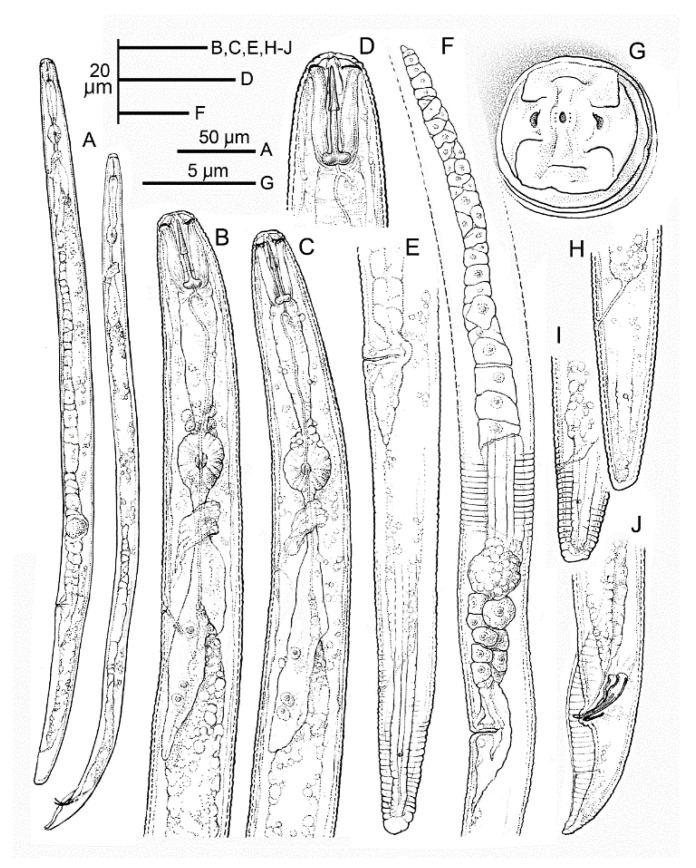
Line drawings of *Pratylenchus vovlasi* sp. nov. (**A**) entire female and male; (**B**–**C**) the female (**B**) and the male (**C**) pharyngeal regions; (**D**) detail of the female lip region; (**E**) detail of the female posterior region; (**F**) the female reproductive system with details of the lateral field; (**G**) en face view; (**H**,**I**) the female tail; (**J**) the male tail.

**Figure 6 plants-10-01068-f006:**
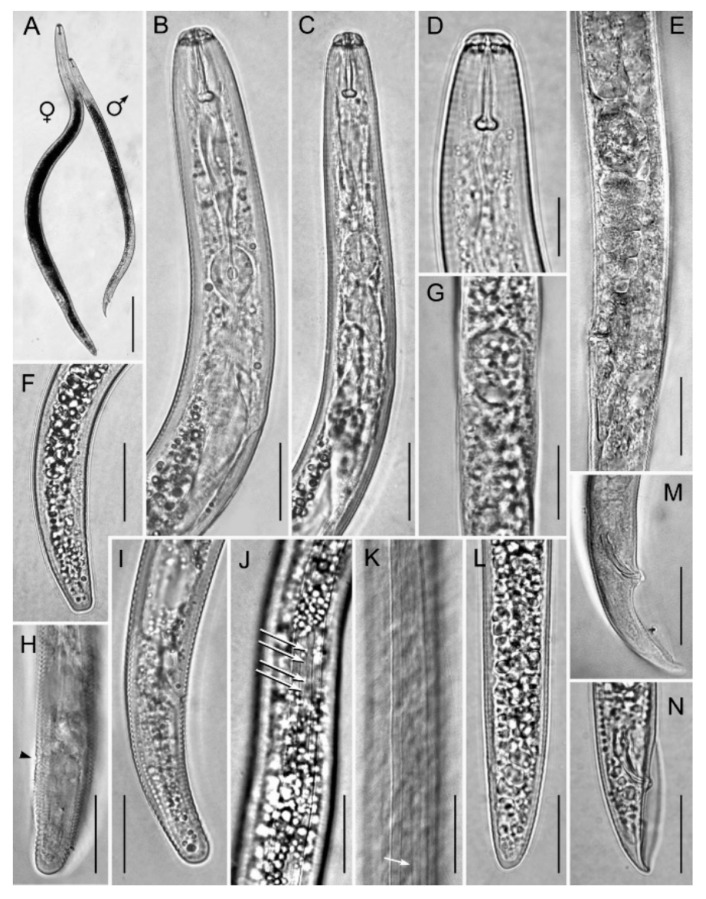
Light photomicrographs of *Pratylenchus vovlasi* sp. nov.: (**A**) entire female and male; the female (**B**) and the male (**C**) anterior regions; (**D**) the female lip region; (**E**,**G**) detail of the vulval region showing the spermatheca; (**F**,**H**,**I**,**L**) detail of the female tail (arrow in H shows the anus); (**J**,**K**), detail of the lateral field incisures showing the oblique striae (arrowed in J) or an additional line in the central band (arrowed in K); (**M**,**N**), detail of the male tail. Scale bars: A = 100 μm; B, C, E–M = 20 μm; D = 10 μm.

**Figure 7 plants-10-01068-f007:**
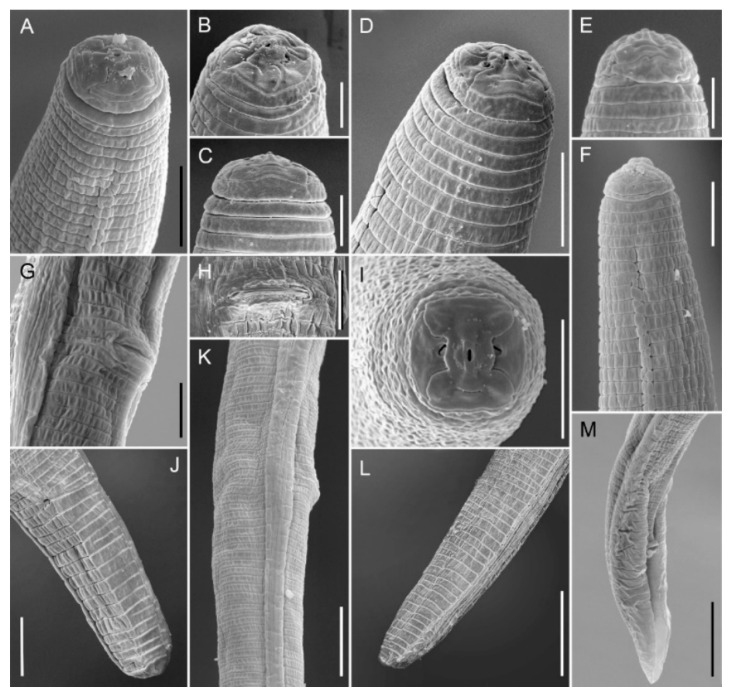
SEM photomicrographs of *Pratylenchus vovlasi* sp. nov.: (**A**–**D**), detail of the female lip region; (**E**,**F**), detail of the male anterior region; (**I**) the female en face view; (**G**,**H**) the vulval region in the sub-lateral (**G**) and the ventral (**H**) view; (**J**–**L**), detail of the female tail; (**M**) detail of the male tail. Scale bars: (**A**,**D**,**F**–**J**) = 5 μm; (**B**,**C**,**E**) = 2.5 μm; (**K**–**M**) = 10 μm.

**Table 1 plants-10-01068-t001:** Morphometrics of the holotype and paratypes of *Pratylenchus vovlasi* sp. nov. All measurements are in µm and in the form: mean ± S.D. (range).

Character	Holotype	Paratypes	
		Females	Males
*n*	-	16	9
L	463	510 ± 33.4 (459–577)	481 ± 32.1 (411–518)
a	21	23.8 ± 2.2 (20.0–26.5)	28.3 ± 1.8 (26.0–31.9)
b	5.8	6.5 ± 0.5 (5.8–7.3)	6.0 ± 0.3 (5.6–6.4)
b’	4.4	4.7 ± 0.4 (4.3–5.5)	4.5 ± 0.2 (4.1–4.9)
c	22.4	22.3 ± 2.7 (18.2–27.2)	17.9 ± 4.0 (9.6–22.0)
c’	1.7	1.8 ± 0.2 (1.4–2.3)	2.6 ± 0.7 (2.0–4.3)
Lip region height	2.0	2.3 ± 0.3 (2.0–2.8)	2.3 ± 0.2 (2.0–2.7)
Lip region diameter	8.5	8.1 ± 1.0 (6.6–9.0)	7.4 ± 0.6 (6.3–8.0)
Stylet length	14.5	15.0 ± 0.6 (14.3–16.3)	14.0 ± 0.6 (13.5–15.0)
Stylet cone	7.0	6.8 ± 0.6 (6.0–8.0)	6.5 ± 0.4 (6.5–7.5)
Stylet knob width	4.0	4.0 ± 0.6 (3.0–4.5)	2.7 ± 0.4 (2.5–3.5)
DGO from stylet base	2.0	2.3 ± 0.3 (2.0–2.7)	1.7 ± 0.4 (1.5–2.5)
o	13.8	15.2 ± 1.9 (13.6–18.4)	11.7 ± 2.8 (9.9–16.3)
**Anterior End to:**			
center of metacorpus	48.0	54 ± 2.6 (48–58)	52.0 ± 2.8 (47.5–55.5)
cardia	80.0	79 ± 5.4 (70–85)	80.0 ± 5.1 (73.5–90.0)
end of pharyngeal lobe	105.0	110 ± 9.0 (97–125)	108 ± 6.5 (99.5–120)
secretory/excretory pore	75	79 ± 3.5 (72–84)	77.5 ± 5.6 (65.5–83)
vulva	347	397 ± 31.8 (347–462)	-
Pharyngeal overlap	25	32 ± 6.7 (20–43)	28.0 ± 4.3 (22.5–36.5)
Max body diameter	22.0	22.0 ± 1.5 (20–24.5)	17.0 ± 1.0 (15.5–18.5)
Anal body diameter	12.0	13.0 ± 1.1 (11.5–15.0)	11.0 ± 1.0 (9.5–12.5)
Anterior genital tract length	224.0	250 ± 30.8 (203–313)	192 ± 23 (163–232)
Spermatheca to vagina distance	42.0	50.0 ± 9.6 (36.5–64.5)	-
Spermatheca length	17.0	17.0 ± 2.5 (12.5–21.0)	-
Spermatheca width	14.0	15.0 ± 1.9 (11.5–18.0)	-
Vulva to anus distance	96	90.0 ± 10.0 (78.5–110)	-
V or T	75	77.8 ± 2.0 (74–80)	40.1 ± 5.0 (32–50)
G1	48	49.0 ± 6.1 (39–59)	-
PUS	26	21.7 ± 2.4 (18.0–26.0)	-
Tail length	20.7	23.5 ± 3.3 (18.0–30.0)	26.5 ± 4.6 (19.5–31.5)
Number of tail annuli	16	16 ± 2.0 (14–20)	-
Spicule length	-	-	16.5 ± 1.4 (14.5–18.5)
Gubernaculum length	-	-	5.0 ± 0.3 (5.6–6.4)

Abbreviations: a = body length/greatest body diameter; b = body length/distance from the anterior end to the pharyngo-intestinal junction; DGO = distance between the stylet base and the orifice of the dorsal pharyngeal gland; c = body length/tail length; c’ = tail length/tail diameter at the anus or cloaca; G1 = anterior genital branch length expressed as a percentage (%) of the body length; L = overall body length; *n* = number of specimens on which measurements are based; o = distance from the stylet base to the dorsal esophageal gland outlet × 100/total stylet length; T = distance from the cloacal aperture to the anterior end of the testis expressed as a percentage (%) of the body length; V = distance from the body anterior end to the vulva expressed as a percentage (%) of the body length; PUS = Post uterine sac length.

**Table 2 plants-10-01068-t002:** Polytomous key of *Pratylenchus vovlasi* sp. nov. and the morphologically most closely related species.

Species	Morphological Characteristics *										
	A_1–3_	B_1–2_	C_1–5_	D_1–4_	E_1–4_	F_1–6_	G_1–3_	H_1–4_	I_1–4_	J_1–3_	K_1–2_
	Lip Annuli	Presence of Males	Stylet Length (µm)	Shape of Spermatheca	Vulva Position (%)	PUS ** (µm)	Female Tail Shape	Female Tail Tip	Pharyngeal Overlap (µm)	Lateral Field	Lateral Field Structures
***Pratylenchus vovlasi* sp. nov.**	**A2**	**B2**	**C2**	**D2**	**E2**	**F3**	**G1,2**	**H1**	**I2**	**J1**	**K1**
*Pratylenchus bhattii*	A2	B2	C2	D2	E1	F2	G2	H1	I2	J1	K1
*Pratylenchus mediterraneus*	A2	B2	C2	D2	E2	F3	G2	H1	I3	J1	K1
*Pratylenchus kralli*	A2	B2	C2	D2	E2	F1	G3	H1	I1	J1	K1
*Pratylenchus fallax*	A2	B2	C3	D2	E2	F3	G3	H2	I2	J1	K1
*Pratylenchus convallariae*	A2	B2	C3	D2	E2	F6	G2	H2	I3	J1	K1
*Pratylenchus penetrans*	A2	B2	C3	D2	E3	F4	G2	H1	I3	J1	K1
*Pratylenchus pratensis*	A2	B2	C2	D4	E2	F3	G3	H2	I1	J1	K1
*Pratylenchus pseudopratensis*	A2	B2	C3	D4	E2	F3	G3	H2	I3	J1	K1
*Pratylenchus thornei*	A2	B2	C2	D2	E2	F3	G2	H1	I3	J1	K1
*Pratylenchus vulnus*	A2	B2	C2	D3	E2	F6	G3	H3	I2	J1	K1

* Morphological characteristics according to Castillo and Vovlas [12]. **Group A: 1** = two; **2** = three; **3** = four. **Group B: 1** = absent; **2** = present. **Group C: 1** = <13; **2 =** 13–15.9; **3** = 16–17.9; **4** = 18–20; **5** = >20. **Group D: 1** = absent or reduced; **2** = rounded to spherical; **3** = oval; **4** = rectangular. **Group E: 1** = <75; **2** = 75–79.9; **3** = 80–85; **4** = >85. **Group F: 1** = <16; **2** = 16–19.9; **3** = 20–24.9; **4** = 25–29.9; **5** = 30–35; **6** = >35. **Group G: 1** = cylindrical; **2** = subcylindrical; **3** = conoid. **Group H: 1** = smooth; **2** = striated; **3** = pointed; **4** = with ventral projection. **Group I: 1** = <30; **2** = 30–39.9; **3** = 40–50; **4** = >50. **Group J: 1** = four; **2** = five; **3** = six to eight. **Group K: 1** = smooth bands; **2** = partially or completely areolated bands. ** PUS = post-uterine sac length.

## Data Availability

Data is contained within the article.

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
