# Peer review of "Pratylenchus vovlasi sp. Nov. (Nematoda: Pratylenchidae) on Raspberries in North Italy with a Morphometrical and Molecular Characterization†"

_plants, 2021, doi:10.3390/plants10061068_

Round 1

Reviewer 1 Report

The article is dedicated to very important subject - root lesion nematodes associated with raspberries. The research is aimed to delineate the new isolated species of Pratylenchus spp. from Piedmont region (North Italy). The new species was characterized by morphological and molecular approach. The study demonstrates that the application of rRNA molecular markers integrated with morphological description can help in the diagnosis and characterization of root-lesion nematode species. New species of Pratylenchus vovlasi was identified based on D2-D3 expansion domains of 28S rRNA gene, ITS region, and the partial hsp90 gene. In morphometry and morphology this species similar to P. mediterraneus, P. thornei, P. penetrans, P. fallax and P. convallariae. The further research is needed to characterize P.vovlasi. It is very actual research article which is recommended for publication in Plants.

Author Response

We thank a lot reviewer 1 for his revision and suggestions.

Reviewer 2 Report

I consider the manuscript as very well prepared. It deals with very important plant parasitic genus attacking field crops including raspberries. According to my opinion the paper could be published almost in present form.

I have only one remark; you are mentioning close phylogenetic relationship of the P. vovlas n. sp. with P. pseudopratensis, it would be therefore good to include P. pseudopratensis to morphological comparison with P. vovlas (part 2.4.4.).

Author Response

We thank a lot reviewer 2 the revision. As suggested by referee 2, we included in section 2.4.4 the comparison with P. pseudopratensis as well as we added a line in the tabular key (Table 2) with the codes of this species.

Reviewer 3 Report

This manuscript brings to light a report on the occurrence of root-lesion nematodes associated with raspberry fields in Piedmont region, Italy. Two species were detected, Pratylenchus crenatus and a new species identified as P. vovlasi sp. nov. The identification of this novel species was based on the molecular characterization, using D2-D3 expansion domains of 28S rDNA, ITS region, and the partial hsp90 gene, and on very detailed and complete morphological and morphometric studies. Above all, the authors pointed out the differences in relation to the closest species, with great significance and interest for later studies involving RLN. The drawings are superb and the microphotographs are also very illustrative.

This new finding is relevant and represents an additional confirmation of the cryptic diversity of Pratylenchus nematodes. My only concern relies on the fact that in the frame of a survey of plant-parasitic nematodes associated with decaying raspberries (Rubus sp.), in northern Italy, the authors did not provide the number of nematodes found in soil samples. This information would be a good contribution in particular because P. vovlasi sp. nov. is identified as the dominant species.  

For the rest, no incongruence was detected along the text. The Abstract is a bit too detailed and I would recommend removing some parts. The conclusions are consistent with the evidence and arguments presented.

In my opinion, this work is robust, complete, well supported by a significant list of references and constitutes a major contribution to all of those dealing with RLN. The manuscript has very relevant and well organized information and deserves to be spread. I've found no limitations in this work and I have only a few minor comments along the text.

Dear authors,

I congratulate you for this study and I wish you good work and stay safe!

Author Response

We thank a lot reviewer 3. We accepted all suggestions and included all information directly in the text.

Reviewer 4 Report

This paper on a new Pratylenchus is complete and well-illustrated. I always enjoy Dr. Troccoli's beautiful drawings. My question with the paper revolves around the very detailed effort to demonstrate that this is not P. pratensis. The characters given in Table 2 that separate these species don't have a lot of taxonomic value, and the perceived tail shape differences don't appear to be very strong; of the 10 tail shapes shown in Figs. 5-7, there is only one clearly truncate tail tip. Many of the morphometric differences given among the species have significant overlaps and are also of small value. A very slight overlap is tolerable but a significant overlap doesn't work. Finally, spermatheca shape is difficult to categorize; I have found trends among species but an ironclad shape for a particular species is unlikely.

I am not that well-posted on molecular analysis, but I'm a bit concerned about the acceptance of 96-97% similarity as being enough for this species, whereas the hsp90 split into two isoforms is dismissed. The authors are probably correct in their analysis but I would like to see it explained a little better.

I've made quite a few suggestions for clarity; but in general the paper is well-written. The minor revision is checked below, although I think it needs more than that. However, I do not see a need for reconsideration if the issues suggested in this review are considered and accepted or rejected with reasonable argument.

Author Response

We accepted all suggestions proposed by reviewer 4 and included all information directly in the text. Specific comments to some points raised by rev 4 follows:

D3: we have added the number of specimens amplified and clarify the number of individual and clones for the hsp90 gene.

D4: we have tried to explain why this gene is important in Pratylenchus species.

D5: figure numbers in descriptions were included, in particular with tail shape.

D6: we agree with rev 4 but we think that citation of literature in this section is functional to better understand patterns which we refer to.

D9: we provided information concerning the locality where the isolate were collected from, as well as the number of nematodes recovered.

D12: regarding Fig. 7J, we want to point out that the photo shows the female tail in ventral view not in lateral and, therefore, lateral fields are not visible. Moreover, we added some references to the figures as suggested.

D13-D14: concerning the comments on tail tip shape and variability we would like to point out that the original isolate collected from raspberries showed typically truncate tail tips, whereas isolates reared on carrot disc cultures showed a wider variability, including tail tips with more rounded margins or with conical to coarsely pointed or indented, striated termini.

D15-17: we agree with the rev 4’ comments, although it is well known from literature that nematodes species belonging to Pratylenchus genus show very similar morphology thus showing significant overlap in most diagnostic characters.